# Important End-of-Life Topics among Latino Patients and Caregivers Coping with Advanced Cancer

**DOI:** 10.3390/ijerph19158967

**Published:** 2022-07-23

**Authors:** Lianel Rosario-Ramos, Keishliany Suarez, Cristina Peña-Vargas, Yoamy Toro-Morales, Rosario Costas-Muñiz, Eida Castro, Stephanie Torres, Normarie Torres-Blasco

**Affiliations:** 1School of Behavioral and Brain Sciences, Ponce Health Sciences University, Ponce 00716, Puerto Rico; lrosario21@stu.psm.edu (L.R.-R.); ksuarez21@stu.psm.edu (K.S.); ytoro19@stu.psm.edu (Y.T.-M.); ecastro@psm.edu (E.C.); storres21@stu.psm.edu (S.T.); 2Ponce’s Research Institute, Ponce Health Sciences University, Ponce 00716, Puerto Rico; cpena@psm.edu; 3Memorial Sloan-Kettering Cancer Center, Department of Psychiatry & Behavioral Sciences, New York, NY 10065, USA; costasmr@mskcc.org

**Keywords:** advanced cancer, end-of-life, patients, caregivers, topics, palliative

## Abstract

There is a known end-of-life related disparity among Latino individuals, and there is a need to develop culturally sensitive interventions to help patients and caregivers cope with advanced cancer. Latino patients and caregivers coping with advanced cancer were asked to list important end-of-life topics to culturally inform a psychosocial intervention adaptation process. A qualitative study was conducted, and semi-structured interviews were performed, audio-recorded, and transcribed. Recordings and transcriptions were reviewed and analyzed using thematic content coding. The semi-structured interview described and demonstrated intervention components and elicited feedback about each. Free listing method was used to assess important topics among Latino advanced cancer patients (*n* = 14; stage III and IV) and caregivers. Patients and caregivers were given a list of 15 topics and asked which topics they deemed important to integrate into the intervention. Overall, more than half of the participants considered it important to include 13 of the 15 topics related to daily activities (eight subcategories), psychosocial support (three subcategories), discussing diagnosis and support (three subcategories), and financial difficulties (one subcategory). Patient-caregivers reported importance in most end-of-life topics. Future research and intervention development should include topics related to psychosocial support, daily activities, discussing diagnosis and support, and financial difficulties.

## 1. Introduction

The Latino disparity in access to health care includes late advanced cancer diagnosis [1], barriers in access to care [2,3,4,5], and unknown cultural needs [3]. This disparity has contributed to a gap in the literature regarding what themes or topics to discuss during end-of-life care among Latino advanced cancer patients and caregivers [6,7]. This translates to a lack of culturally sensitive interventions within this population. Because Latino cultural norms [4,8] and attitudes [5] influence end-of-life care, exploring important topics during the end-of-life stage would help to provide personalized care. 

Determining and providing patients with end-of-life topics may provide a bridge that can assist in discussing difficult themes during this stage [9]. Because of Latino culture, cancer patients would rather not talk directly about the end-of-life process [10]. Discussing important end-of-life themes among Latino patients may serve as a facilitator in many areas. Related to healthcare, it can assist in treatment decision-making [11,12,13,14], improve end-of-life care [15], and improve the general quality of life [16]. These topics could also serve as facilitators during the dying process and preparation for death [14,17,18]. Regarding caregivers and family members, it can help to improve communication with family members and caregivers [11,12,18], minimize distress and anxiety while discussing difficult topics [19], and reduce psychological distress in grieving caregivers and family members [20]. 

Latino culture has cultural values that influence health behaviors and cultural aspects of care [21]. The literature suggests that some of these cultural values include *familismo, machismo, and marianismo. Familismo* describes the importance and loyalty to family [22] and how one prioritizes the family over individual interests [23,24]. Machismo revolves around a society of pride and the expectation of being a man, which includes protection, manliness, and providing for the family [25]. Marianismo is the cultural norm where the woman is viewed as a caregiver [25] and submissive [26]. By addressing important topics and cultural norms, such as *familismo, marianismo, and machismo*, we may know culturally sensitive topics to address the end-of-life needs, which include psychosocial care, stigma [27], finances [28], and the subject of death and the process of dying [29]. Culturally sensitive interventions should consider factors and cultural aspects of everyday life [30].

According to the literature, there is a gap in identifying topics relating to Latino coping with advanced cancer [6,7]. Therefore, it is essential to understand the most salient themes among patients and caregivers so we can help them improve their emotional wellbeing, reduce their burden, and provide tools to aid in the end-of-life process [7]. This article aims to describe important Latino end-of-life topics in a psychosocial intervention for patient-caregivers coping with cancer to inform the adaptation process culturally.

## 2. Materials and Methods

Participants were recruited from an oncology clinic in the southern area of Puerto Rico between October 2020 and September 2021. The Ponce Research Institute Institutional Review Board (IRB) approved all the study procedures. An IRB-approved introductory letter introduced potential participants to the study. Patient-caregivers’ inclusion criteria were to be Latino (a person born in Latin America) and obtain a score of >3 on the Distress Thermometer. The Distress Thermometer is a tool used to screen potential distress and symptoms among cancer patients [31]. An in-person research staff member followed up with potential participants to provide information, answer questions, and determine eligibility, including the Distress Thermometer. Those eligible and interested completed informed consent. For those who gave consent, a call was scheduled to complete the questionnaire and interview.

We conducted a free listing study to systematically assess important end-of-life topics among Latino patients and caregivers coping with advanced cancer. Free listing is a standard anthropological method used to define domain elements and measure the extent to which group members share those definitions [32,33]. The method has previously been used to effectively compare disease notions and perceptions of health practices from the standpoint of health care providers and consumers in several contexts, including pediatric head injury management, depression, and ADHD [34,35,36,37,38].

Between October 2020 and September 2021, we conducted semi-structured interviews and free listing questions with a convenience sample of patients and caregivers recruited from an oncology clinic in the southern area of Puerto Rico. Due to the ongoing COVID-19 pandemic, interviews were conducted via Zoom calls to protect the patients’ and caregivers’ welfare. Patients and caregivers were interviewed separately to avoid any influence on their answers. Participants were given USD 30 as compensation for their time and effort. Patients’ eligibility was determined according to the following parameters: over 21 years of age and meeting the criteria for an advanced cancer stage (stage III or IV). Eligible caregivers were parents or relatives of patients who accompanied the enrolled patient to their appointments and were involved with the patient’s care (e.g., taking part in treatment meetings with the patient). According to prior literature, qualitative exercises ideally require 12 participants or until saturation occurs [39]. We recruited 14 dyads (patients and caregivers) until saturation was met with the free listing. All participants were given the opportunity to decline participation or terminate the free listing questions at any time. Each participant was provided written informed consent. For patients, this included granting the researchers permission to access electronic medical records for diagnostic information. All individuals invited to participate agreed to do so.

The mentor (N.T.-B.) conducted all free list interviews in person in a private room at the oncology clinic. A brief questionnaire collected basic demographic data, including age, gender, race/ ethnicity, and years of education for patients and caregivers. In addition, we recorded the caregiver’s relationship to the patient (mother, father, other) and, for clinicians, the number of years as a psychiatric service provider. Finally, participants completed free listing questions that lasted approximately five minutes. During the interview, each participant was instructed to talk about all the important topics that came to mind in response to the most important topic to address after the cancer diagnosis.

The qualitative codes used for the interviews include the themes: “reaction to diagnosis”, “dealing with changes in your physical appearance caused by cancer treatment”, “the importance of talking about diagnosis and receiving support by getting support from friends and family”, “disruptions to your life caused by medical appointments or hospitalizations”, “having to give up or cut back from work or other family activities”, “plans for the future”, “fear or worries about death”, “maintaining a satisfactory sex life”, “completing household tasks”, “disruptions to your life caused by disease side effects”, “telling your friends or family members about the illness”, “talking with children about cancer”, “financial difficulties”, “being hospitalized”, and “difficulties completing daily tasks”.

The theme was noted as coded when the patient clearly stated that he or she liked to include the theme. When the patients did not report the topic, it was not coded in the free listing. Using ATLAS.ti’s report and query functions, the qualitative analysts (C.P.-V., L.R.-R., and K.S.) independently coded the free listing [40,41,42,43]. The qualitative coders then coded the remaining transcriptions using the free listing, and meetings were held to reach a consensus about applied codes. Through consensus meetings, divergence and convergence points were discussed amongst the group until consensus was met. There were no major differences between the coders, and, if there was a minor difference (two to one), a consensus was reached after discussion. Reliability was conducted through team-based consensus building. The team’s previous publications include more details about the methodology [42,43,44,45]. All investigators had expertise in qualitative analysis, and the last author moderated these discussions [42,43,44,45].

Word lists were first examined, and plural and singular terms and synonymous words were combined. For example, terms such as “tratamiento” and “tratamientos” were both categorized as “treatment”. “Diagnóstico” and “diagnósticos” were both categorized as “diagnostics”. Two research team members reviewed all free list data to ensure reliability and standardized word formatting when combining synonyms. Reviewers were blind to any information concerning the participants’ study group when reviewing and combining terms. Data were then imported into ATLAS.ti and analyzed by deductive content analysis.

## 3. Results

### 3.1. Sociodemographic Information

Seven patient–caregiver dyads completed the interview, for a total of 14 participants. The patients’ age mean was 59 years (SD = 11, range = 40–76), 57% were female, and 100% were Latino. Patients’ diagnoses included stage III of cancer (*n* = 3) and stage IV of cancer (*n* = 10). The types of cancer included: breast cancer (*n* = 3), multiple myeloma (*n* = 2), cervical cancer (*n* = 1), head and neck cancer (*n* = 1), lung cancer (*n* = 1), malignant carcinoid (*n* = 1), multiple myeloma (*n* = 1), non-cell lung cancer (*n* = 1), prostate cancer (*n* = 1), small lung cancer (*n* = 1), and squamous cell carcinoma (*n* = 1). Caregivers’ mean age was 52 years (SD = 13, range = 25–79), 57% were female, and 100% were Latino. Nine caregivers were spouses, two sisters, one daughter, one grandson, and one friend. The dyadic free list important topics average ranged from 85% to 21%, patients’ important topics average ranged from 71% to 28%, and caregivers’ average of important topics ranged from 100% to 42%.

### 3.2. Salient Themes to Be Discussed in End-of-Life by Dyads

The salient terms shared in response to the questions are organized by respondent type and summarized in Table 1. Frequency reflects the number of times a term was stated among the group in response to a question. Average (%) rank reflects the position in which a term appears on a list and is averaged across all the participants in that group.

When asked what important topics to talk about after the cancer diagnosis, patients and caregivers (*n* = 12) chose “reaction to diagnosis” and “dealing with changes in your physical appearance caused by cancer treatment”, which were the salient topics shared across the two groups. Patients and caregivers (*n* = 11) expressed the importance of “receiving support from friends and family”. Patients and caregivers (*n* = 10) deemed it important to discuss “disruptions to your life caused by medical appointments or hospitalizations” and “having to give up or cut back from work or other important activities”. When asked, participants (*n* = 9) chose “plans for the future”, “fear or worries about death”, “maintaining a satisfactory sex life”, and “completing household tasks”. Participants (*n* = 8) selected “disruptions to your life caused by disease side effects”, “telling friends or family members about the illness”, “talking with children about cancer”, and “financial difficulties”. Patients and caregivers (*n* = 7) selected “being hospitalized” as an important topic to discuss. Finally, the topic of most minor importance for patients and caregivers (*n* = 3) was “difficulties completing daily activities”.

### 3.3. Salient Themes to Be Discussed in End-of-Life Stage by Patient

The patients listed 15 important topics to discuss after a cancer diagnosis. Patient responses are listed in Table 2. When asked, almost all the patients (*n* = 6) chose “maintaining a satisfactory sex life” as an important topic. Patients (*n* = 5) selected the topics “completing household tasks”, “dealing with changes in physical appearance caused by cancer treatment”, and “dealing with changes in physical appearance caused by cancer treatment”. When prompted, patients (*n* = 4) picked “receiving support from friends and families”, “having to give up or cut back from work or other important activities”, and “fear or worries about death”. Patients (*n* = 3) chose “disruptions to your life caused by medical appointments or hospitalizations”, “plans for the future”, “disruptions to your life caused by disease side effects”, “telling friends or family members about the illness, and “talking with children about cancer”. As the second least favorite topic, patients (*n* = 2) selected “financial difficulties” and being hospitalized”. Finally, the least favorite topic among patients was “difficulties completing daily activities (*n* = 0).

### 3.4. Salient Themes to Be Discussed in End-of-Life Stage by Caregivers

The caregivers listed important topics to discuss after a cancer diagnosis. Caregiver responses are presented in Table 3. All the caregivers (*n* = 7) expressed the importance of discussing “reaction to diagnosis”, “dealing with changes in physical appearance caused by cancer treatment”, “receiving support from friends and families”, and “disruptions to your life caused by medical appointments or hospitalizations”. When asked, caregivers (*n* = 6) chose “Having to give up or cut back from work or other important activities”, “Plans for the future”, and “financial difficulties”. Caregivers (*n* = 5) found the following topics important, “fear or worries about death”, “disruptions to your life caused by disease side effects”, “telling friends or family members about the illness”, “talking with children about cancer”, and “being hospitalized”. Some caregivers (*n* = 4) found the topic of “completing household tasks” important. Finally, caregivers (*n* = 3) found “maintaining a satisfactory sex life”, and “difficulties completing daily activities” the least favorite topics. 

## 4. Discussion

This paper describes important end-of-life topics among Latino patient–caregiver dyads coping with advanced cancer. A free listing method was conducted to discover important end-of-life topics to discuss during the end-of-life stage after receiving a cancer diagnosis. Fifteen topics were equally presented to both groups (patients and caregivers). The findings show that, out of the 15 topics, 14 topics were selected from both groups. The most shared salient topics were “reaction to diagnosis” and “dealing with changes in your physical appearance caused by cancer treatment”. However, the patient–caregiver dyads found “difficulties in daily activities” to be the least important topic of discussion, chosen by three caregivers and zero patients. The results suggest that patients and caregivers would benefit from including essential topics during a psychosocial intervention, said topics being: “reaction to diagnosis”, “dealing with changes in your physical appearance caused by cancer treatment”, “the importance of talking about diagnosis and receiving support by getting support from friends and family”, “disruptions to your life caused by medical appointments or hospitalizations”, and “having to give up or cut back from work or other family activities”.

It was observed that advanced cancer patients underscored the importance of having effective communication during end-of-life care. The results are comparable to other studies that aimed to understand perceptions [46] and essential elements [47] in advanced cancer patients and families. The results also indicate differences in answers by the dyads; none of the topics had the same number of responses. Almost all the patients selected “maintaining a satisfactory sex life” as the most important topic, while less than half of the caregivers chose this topic. These results are similar to other literature highlighting the importance of the patient’s sexuality in the end-of-life stage [48]. Concerning financial difficulties, our results show a disparity in the need to discuss financial difficulties. Most caregivers believe it is an important topic, whereas patients do not [47]; however, at this stage, it is considered an important topic [49]. This difference could be attributed to the financial burden cancer caregivers experience [50]. Patients and caregivers experience and process the cancer diagnosis and process differently, which could explain why we see a difference in the answers provided [51,52].

As with other investigations, where they highlight the importance of discussing and offering the necessary tools to patients and family members during the end-of-life stage [53], the results yielded from free listing shed light on themes that can aid the end-of-life stage process. The mentor and primary investigator (N.T.-B.) is currently leading a cultural adaption of a psychosocial intervention aimed at Latino patients and caregivers coping with advanced cancer [54]. Said intervention will consider which topics patients and caregivers would like to discuss and integrate them appropriately during their end-of-life stage. Some possible limitations include limited responses and a possible lack of depth that is presented in the free listing. Further exploration is needed to study the specific impact Latino culture and norms have on the selection of important themes and how they affect the end-of-life stage. Moreover, we did not include information on a participant’s nationality.

## 5. Conclusions

In conclusion, more than half of Latino patient–caregiver dyads coping with advanced cancer found most end-of-life topics important. Some topics were considered to be more important by patients, whereas caregivers considered some topics more important. Determining which topics are considered essential for each group facilitates end-of-life or palliative care and can serve as a tool to assist in communication and decision-making. Integrating cultural norms within the Latino community may explain why some discussions are integrated during the end-of-life stage. Presenting what topics patient–caregiver dyads consider essential for discussion can be used to culturally aid a psychosocial adaptation intervention process. Future research should consider the use of the topics presented as a guide for future psychosocial interventions with advanced cancer patients.

## Figures and Tables

**Table 1 ijerph-19-08967-t001:** Dyadic response to end-of-life topics (*n* = 14).

Topics	Response
Reaction to diagnosis	12
Dealing with changes in your physical appearance caused by cancer treatment	12
Importance of talking about diagnosis and receiving support by getting support from friends and family	11
Disruptions to your life caused by medical appointments or hospitalizations	10
Having to give up or cut back from work or other family activities	10
Plans for the future	9
Fear or worries about death	9
Maintaining a satisfactory sex life	9
Completing household tasks	9
Disruptions to your life caused by disease side effects	8
Telling your friends or family members about the illness	8
Talking with children about cancer	8
Financial difficulties	8
Being hospitalized	7
Difficulties completing daily tasks	3

**Table 2 ijerph-19-08967-t002:** Patient response to end-of-life topics (*n* = 7).

Topics	Response
Maintaining a satisfactory sex life	6
Reaction to diagnosis	5
Dealing with changes in your physical appearance caused by cancer treatment	5
Completing household tasks	5
Importance of talking about diagnosis and receiving support by getting support from friends and family	4
Having to give up or cut back from work or other family activities	4
Fear or worries about death	4
Disruptions to your life caused by medical appointments or hospitalizations	3
Plans for the future	3
Disruptions to your life caused by disease side effects	3
Telling your friends or family members about the illness	3
Talking with children about cancer	3
Financial difficulties	2
Being hospitalized	2
Difficulties completing daily tasks	0

**Table 3 ijerph-19-08967-t003:** Caregiver response to end-of-life topics (*n* = 7).

Topics	Response
Reaction to diagnosis	7
Dealing with changes in your physical appearance caused by cancer treatment	7
Importance of talking about diagnosis and receiving support by getting support from friends and family	7
Disruptions to your life caused by medical appointments or hospitalizations	7
Having to give up or cut back from work or other family activities	6
Plans for the future	6
Financial difficulties	6
Fear or worries about death	5
Disruptions to your life caused by disease side effects	5
Telling your friends or family members about the illness	5
Talking with children about cancer	5
Being hospitalized	5
Completing household tasks	4
Maintaining a satisfactory sex life	3
Difficulties completing daily tasks	3

## Data Availability

Data can be made available upon request.

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
