# Peer review of "Important End-of-Life Topics among Latino Patients and Caregivers Coping with Advanced Cancer"

_ijerph, 2022, doi:10.3390/ijerph19158967_

Round 1

Reviewer 1 Report

Dear authors:

At the end of life phase the respect of every individual opinion and story is very important. Communication is usually key and although there's a voiced concern regarding always telling the truth, this should be done wisely, assessing what the patient knows, wants to know and needs to know. The same thing is applied to the family.

Your study searches similarities in a latinx patients and families regarding general topics. Although this is important, is the difference between them that is the most interesting because all of the above stated. I found no reference in your study to this aspect or its importance.  

Furthermore, I would like you to explain how you intend to use this topics  and which precautions you would need to take on using them.

I also find no reference to an ethical committee submission.

Thanks

Author Response

Thank you for your review and recommendations. Responses are attached in a word file. 

Reviewer 2 Report

I have reviewed the manuscript entitled “Important End-of-Life Topics Among Latinx Patients and Caregivers Coping with Advanced Cancer”. The theme developed is very interesting and original. It allows to contribute to the development of a program that could become an interesting tool to support caregivers and patients with advanced cancer. I believe readers of the journal with a special interest in palliative care will appreciate your work as I have.

I can only suggest a clarification on:

a) If exists a code associated with the decision of the ethics committee, please You should insert it;

b) Improve the description of the content analysis process used (what is the framework theory used: phases?)

I have nothing to add and I wish you all the best with its publication. I have nothing to add and I wish you all the best with its publication.

I believe the journal's readers will appreciate your paper.

I have nothing to add and I wish you good luck towards publishing the paper!

Author Response

(The authors gave the same response as above.)

Reviewer 3 Report

This is a prospective, qualitative research based on interviews. The design followed the steps of conventional scientific methodology. The text follows the academic format and describes the ethical precaution of research. The conclusion contributes to the healthcare and quality of the studied public. 

The following demands are for the paper to conform to the standards of this journal. I ask the authors to consider these remarks:

- the period in which the data collection took place was during the covid-19 pandemic; please include one or two phrases about the patient-caregiver relationship at that time, in light of the recommendation for social distancing.

- It was not clear to me whether the interviews were individual or whether they were carried out in the presence of both dyad components together (if this is already in the text, it may have been my fault).

- the requirement of 12 participants for sufficiency (line 91) also refers to the mix of patients and caregivers? Are not they two very different publics? Please comment on "Discussion."

- "Latinx" is a neologism that could not be broadly known worldwide. Furthermore, the term elicits some controversy - https://link.springer.com/article/10.1057/s41276-018-0142-y -. I think some words are necessary on what the term means and why it was chosen.

- At the authors' discretion, the word "sex" could be replaced by "gender."

Author Response

(The authors gave the same response as above.)

Reviewer 4 Report

This article describes results from a qualitative study that employs semi-structured interviews to determine important end-of-life topics among Latinx patients and caregivers coping with advanced cancer.  Authors contend that the end-of-life-related disparity among Latinx patients is known.  Moreover, there is a need to understand topics of importance to Latinx patients and caregivers and devise culturally sensitive interventions to help patients and caregivers cope with advanced cancer.  Authors aim to address this gap in the literature with their study. The review is as follows:

1.     Since the Latinx population is heterogenous group, authors should define for the readers how ‘Latinx’ is defined for their study.  Furthermore, for sociodemographic data, was information collected on nationality of participants?

2.     Good definitions of familismo, machismo and marianismo in the Introduction.

3.     Line 70 – For ‘Patients-caregivers inclusion criteria was to obtain a score of >3 on Distress Thermometer’, is the Distress Thermometer an existing scale?  If so, is there a citation to show its application?  If referring to the distress thermometer for caregivers, authors should define and describe the Distress Thermometer.

4.     For the Materials and Procedures, what were the recruitment methods?  When and where were participants recruited?  Were participants compensated for their time?

5.     The analysis plan is not well-defined, and it is a bit unclear how the results were determined.

6.     Good definition of a free listing study in the Materials and Methods.

7.     For Tables 2 and 3, authors should consider listing the responses in descending order (high to low) to match how the responses are listed in Table 1.

8.     Lines 185-86 – In ‘Results suggest patients and caregivers would benefit from including certain topics during a psychosocial intervention’, this sentence is unclear.  What are the certain topics?

Overall, this is a unique, pertinent, and laudable effort by the authors.  This is an important topic that could address a gap in the literature.  Some areas such as the methods and analysis are a bit unclear.  Addressing these items may help to improve the paper.

Author Response

(The authors gave the same response as above.)

Round 2

Reviewer 4 Report

Authors have sufficiently addressed the requested feedback, including to clarify the recruitment methods, data analysis plan, and definition of the study population.  The revised paper is more detailed and clearer.